# *Achillea millefolium* L. and *Achillea biebersteinii* Afan. Hydroglycolic Extracts–Bioactive Ingredients for Cosmetic Use

**DOI:** 10.3390/molecules25153368

**Published:** 2020-07-24

**Authors:** Katarzyna Gaweł-Bęben, Marcelina Strzępek-Gomółka, Marcin Czop, Zuriyadda Sakipova, Kazimierz Głowniak, Wirginia Kukula-Koch

**Affiliations:** 1Department of Cosmetology, University of Information Technology and Management in Rzeszów, Sucharskiego 2, 35-225 Rzeszów, Poland; mstrzepek@wsiz.rzeszow.pl (M.S.-G.); kglowniak@wsiz.rzeszow.pl (K.G.); 2Department of Clinical Genetics, Medical University of Lublin, Radziwiłłowska 11, 20-080 Lublin, Poland; marcin.czop@umLub.pl; 3School of Pharmacy, Kazakh National Medical University named after S.D. Asfendiyarov (KazNMU), 88 Tole bi street, 050012 Almaty, Kazakhstan; sakipova.z@kaznmu.kz; 4Department of Pharmacognosy, Medical University of Lublin, Chodźki 1, 20-093 Lublin, Poland; virginia.kukula@gmail.com

**Keywords:** *Achillea millefolium*, *Achillea biebersteinii*, tyrosinase, sun protection factor, melanoma, Asteraceae, ESI-mass spectrometry

## Abstract

Studies on hydroglycolic (HG) extracts of *Achillea biebersteinii* (AB)—a less investigated representative of the genus—were performed to determine their potential for cosmetic applications compared to the well-known *Achillea millefolium* (AM). Three types of water:polyethylene glycol extracts (1:1, 4:1, 6:1 *v*/*v*) were obtained from both species and analyzed for their composition by high performance liquid chromatography coupled with mass spectrometry (HPLC-ESI-Q-TOF-MS) and assayed for their biological activities. The study led to the identification of 11 metabolites from different natural product classes with the highest share corresponding to 5-caffeoylquinic acid, axillarin, coumaroylquinic acid isomers and 3-caffeoylquinic acid. The highest antiradical capacity in DPPH and ABTS scavenging assays was shown for HG 4:1 of AB and AM extracts. HG 1:1 extracts from both species inhibited monophenolase and diphenolase activity of tyrosinase, whereas AB HG 4:1 extract showed significant monophenolase inhibition. The highest sun protection factor (SPF) was determined for AM HG 4:1 extract, equal to 14.04 ± 0.17. The AB extracts were cytotoxic for both human keratinocytes HaCaT and A375 melanoma, however HG 1:1 and 4:1 extracts were more cytotoxic for cancer than for noncancerous cells. In conclusion, AB HG 1:1 and 4:1 extracts display significant potential as active cosmetic ingredients.

## 1. Introduction

Plants are a rich source of bioactive compounds widely used due to their healing and cosmetic properties. The health-promoting activities of many plant-derived compounds have been confirmed by in vitro and in vivo studies. The application of botanical ingredients in cosmetic formulations is often based on their traditional usage to treat specific skin conditions. In recent years a growing amount of scientific evidence has confirmed the cosmetic properties of several plant extracts and ingredients, including *Glycyrrhiza glabra*, *Curcuma longa*, *Psoralea corylifolia*, *Cassia tora*, *Punica granatum*, *Centella asiatica*, *Cinnamomum zeylanicum*, *Aloe vera, Calendula officinalis, Moringa oleifera, Morus alba* and *Cistus spp*. [1,2,3,4]

*Achillea millefolium* L. is one of the most commonly known medicinal plants, used for many centuries as a natural remedy to treat wounds, bleeding, headache, inflammation, pains and gastrointestinal disorders. The medicinal properties of the extracts and the components of *A. millefolium* have also been confirmed by a broad range of scientific studies [5]. Extracts, juice, essential oil and distillate from the aerial parts of *A. millefolium* are also common and valuable active ingredients of cosmetics, acting as skin conditioning, antioxidant, shooting and refreshing agents [6]. Recent studies have proved the strong antioxidant, skin lightening [7], wound healing [8], skin rejuvenating [9] and anti-inflammatory [10] potential of *A. millefolium* alcoholic or hydro-alcoholic extracts. Cosmetic ingredients containing *A. millefolium* preparations are also considered as safe [11].

Extracts from other *Achillea* species are not listed as cosmetic ingredients on the EU’s Inventory of Cosmetic Ingredients (CosIng), except for *Achillea asiatica* flower/leaf/stem extract that is used in cosmetic formulations as a hair and skin conditioning ingredient and humectant [6]. However, there is a growing volume of evidence showing that other *Achillea* species may also possess beneficial properties for skin condition and thus might be used as valuable, multifunctional ingredients for cosmetic use.

One such species is *Achillea biebersteinii* Afan. Extracts obtained from the aerial parts of *A. biebersteinii* accelerate wound healing in vitro and in vivo [12]. Hydro-alcoholic extracts from the flowers of *A. biebersteinii* may inhibit scar formation as they downregulate the expression of transforming growth factor β1 (TGF-β1) and upregulate the expression of basic fibroblasts growth factor (bFGF) on gene and protein levels in murine embryonic fibroblasts in vitro [13]. Cosmetic application of *A. biebersteinii* extracts is also supported by their significant antioxidant properties and antimicrobial activity against pathogens causing dermal infections such as *Staphylococcus aureus* and *Pseudomonas aeruginosa* [7,14].

Available data on the biological activity of *A. biebersteinii* and other *Achillea spp*. are based mostly on alcoholic or hydro-alcoholic extracts with limited applications in actual skin care products. From the practical point of view one of the most useful herbal extracts for cosmetic industry are those containing propylene glycol (PEG) as a solvent or co-solvent. PEG is one of the most widely used ingredients in cosmetics and personal care products, including facial cleansers, moisturizers, bath soaps, shampoos and conditioners, deodorants, shaving preparations, and fragrances. It is listed on the CosIng database and its use as a cosmetic ingredient is not restricted in any way according to the general provisions of the Cosmetics Regulation of the European Union [15]. Glycolic or hydro-glycolic plant extracts offer a wide range of additional advantages – water solubility, biocompatibility, stability and low toxicity. Several studies have also proved that PEG may increase the solubility of different natural ingredients, reduce water activity and thus enhance preservative effect. PEG is also used in cosmetic formulations as humectant, viscosity-decreasing agent, solvent, and fragrance ingredient [16].

The aim of this study was a comparative analysis of six hydroglycolic extracts from the flowering aerial parts of *A. millefolium* and *A. biebersteinii* collected in Kazakhstan, prepared using various water to PEG ratios (1:1, 4:1 and 6:1, *v*/*v*) by a high performance liquid chromatography coupled with mass spectrometry (HPLC-ESI-Q-TOF-MS) approach. The extracts were also compared for their content of chosen phytochemicals and biological properties, such as antioxidant properties, tyrosinase inhibitory activity, in vitro cytotoxicity and sun protection factor (SPF). The obtained results have proved that hydro-glycolic extracts from *Achillea* species, especially *A. biebersteinii*, are highly valuable potential active ingredients for cosmetic formulations.

## 2. Results and Discussion

### 2.1. Composition of the Investigated Samples

The HPLC–ESI-Q-TOF-MS analysis allowed the identification of 11 major secondary metabolites in the analyzed hydroglycolic extracts from *A. millefolium* and *A. biebersteinii*. The compounds found in *A. biebersteinii* extracts included phenolic acids like chlorogenic acids (3-caffeoylquinic acid: 3-CQA, 4-caffeoylquinic acid: 4-CQA and 5-caffeoylquinic acid: 5-CQA), caffeic acid, cynarin (1,3-dicaffeoylquinic acid, 1,3-DCQA), quinic acid, some esters and aldehydes of phenolic acids like coumaroylquinic acid isomers and flavonoids: kaempferol, jaceidin and axillarin (Table 1, Appendix A). In *A. millefolium* extracts phenolic acids like 3-CQA, 4-CQA, 5-CQA, cynarin, quinic acid, esters and aldehydes of phenolic acids like coumaroylquinic acid isomers and flavonoids like kaempferol and jaceidin were identified (Table 2, Appendix A). Several surveys on the chemical composition of different species of *Achillea* can be found in the scientific literature. Zengin and co-investigators in their review [7] described the compositional data acquired by HPLC-MS for three *Achillea* species: *A. millefolium, A.biebersteinii* and *A. teretifolia*. Their results showed the presence of protocatechuic acid, 3-caffeoylquinic acid, 4-caffeoylquinic acid, 1-feruloylquinic acid, 4-feruloylquinic acid, 1,3-dicaffeoylquinic acid, 3,5-dicaffeoylquinic acid, 3,4-dicaffeoylquinic acid and 3,4,5- tricaffeoyl- quinic acid in their extracts. The phytochemical composition of *A. biebersteinii*, *A.setacea* and *A. willhelmsii* were described by Şabanoğlu S. et al. using RP-HPLC-DAD. These studies showed the presence of chlorogenic acid, caffeic acid, rutin, quercetin, luteolin and apigenin in these species [17]. Ali and co-investigators [5] summarized the chemical constituents and phytochemical properties of *Achillea millefolium*. In their paper, 3,5-DCQA, 1,5-DCQA, 4,5-DCQA, neochlorogenic acid and ferulic acid were described as strong antioxidants [1]. Also, Lee et al. [18] examined the antioxidant and antimelanogenic properties of *Achillea alpina* and performed the isolation of the secondary metabolites from this species. They isolated 17 compounds: jaceidin, axillarin, 5,7,4-trihydroxy- 3,6-dimethoxyflavonone, isoshaftoside, shaftoside, neoshaftoside, isovitexin, quercetin, chlorogenic acid, chlorogenic acid methyl ester, 5-*O*-coumaroylquinic acid, 5-*O*-coumaroylquinic acid methyl ester, 3,5-DCQA acid, methyl 3,5-DCQA acid, penduletin, chrysosplenol B, 3-*O*-vicianoside, and achilline. Among the isolates methyl 3,5-DCQA exhibited the most beneficial antioxidant properties and isovitexin significantly downregulated the activity of tyrosinase enzyme, that is crucial in the synthesis of melanin in the skin tissue [14]. Afshari and collaborators [19] performed HPLC analysis of six species of *Achillea*: *A. santolina, A. milelfolium, A. aucheri, A. nobilis, A. filipendulina and A. pachycepchala*. In *A. aucheri, A. nobilis, A. filipendulina* and *A. pachycepchala* and identified gallic acid, chlorogenic acid, caffeic acid, *p*-coumaric acid, rutin, ferulic acid, luteolin-7-*O*-glycoside, 1,3-DCQA, luteolin, quercetin and apigenin in their extracts. In *A. millefolium* according to the authors the metabolites were alike, except from p-coumaric acid and quercetin that were not present. In *A. santolina* extracts quercetin was also not determined [19]. Based on the above examples of the recent studies on different representatives of *Achillea* genus it can be concluded that the majority of the compounds identified in the presented study was also confirmed in other species. In the qualitative composition study described above *A. millefolium* was found to deliver less rich extracts in relation to the other species. The HPLC-ESI-Q-TOF-MS analysis of the obtained extracts of *A. millefolium* does not confirm the presence of caffeic acid and axillarin that were found in *A. biebersteinii*. The lack of a free form of the former compound was in opposition to the previously reported compositional studies performed by Güneş and colleagues [20].

The quantitative study showed a relationship between the content of secondary metabolites and the solvent used for extraction. Based on the previous scientific publications, e.g., [24] HG extracts were found to extract higher quantity of polyphenols from the powdered plant material that was confirmed by their IC_50_ values measurements in the DPPH antiradical assay and the total phenolic content determination. Also, according to the same authors, the percentage content of PEG played a crucial role in the extraction efficiency. The lower the content of water in the mixture, the richer the extracts were. In our studies a mixture of water and PEG (1:1 *v*/*v*) resulted in the richest extracts, whereas the 6:1 (*v*/*v*) hydroglycolic (HG) extracts showed the smallest quantity and variety of components in both tested species.

The leading components identified in *A. biebersteinii* HG 1:1 were 5-CQA (3.181%), axillarin (2.539%), 3-CQA (1.922%) and 4-CQA (1.207%). Similarly HG 1:1 (*v*/*v*) extracts were found to be the most rich in metabolites of *A. millefollium.* Its HG 1:1 extract contained coumaroylquinic acid isomers (0.734 and 1.733%) as the most prominent natural products. The following compounds were confirmed in the same extract: quinic acid, 5-CQA, 3-CQA, 4-CQA, jaceidin, kaempferol and cynarin and their content was equal to: 1.172 ± 0.142%, 0.993 ± 0.092%, 0.595 ± 0.068%, 0.466 ± 0.013%, 0.043 ± 0.008%, 0.272 ± 0.021% and 0.119 ± 0.002%, respectively. *A. millefolium* HG 6:1 extract showed the least beneficial composition. In general, an increasing quantity of water in the HG 6:1 extracts were incapable of the recovery of large quantities of natural products from plant matrix.

In the case of *A. biebersteinii* HG 1:1 extract the highest concentration was determined for 5-DCQA and in *A. millefolium* HG 1:1 extract delivered the highest quantity of coumaroylquinic acid isomers. Both, *A. biebersteinii* and *A. millefolium* contained marked quantities of phenolic acids. In *A. biebersteinii* HG 1:1 extract flavonoids are the second group of compounds and coumaroylquinic acid isomers are the last class of metabolites in the terms of percentage content. In *A. millefolium* extracts there are more coumaroylquinic acid isomers confirmed in contrast to the flavonoids content (Table 3, Appendix A). Recent studies showed extracts rich in phenolic acids and flavonoids are important ingredients of after-sun cosmetics through reduction of oxidative stress, inflammation and immunosuppresion in the UV-exposed skin [25].

A quantitative composition assessment was also performed by Zengin and colleagues on *Achillea phrygia* extracts [23] and Agar et al. [21] on Turkish *Achillea spp*. According to both groups, the maximum quantity of chlorogenic acid in the extract was calculated as 2088 ± 16 or 2890 ± 141.6 μg/g of extract. The herein presented results determine the content of metabolites of interest in the crude plant material and in different extracts that is why it is difficult to directly relate to the previously published results.

### 2.2. The Antiradical Potential of A. Millefolium and A. Biebersteinii Hydro-Glycolic Extracts

The antioxidant activity of hydroglycolic *A. millefolium* and *A. biebersteinii* extracts was not described in the scientific literature to date. In our study the antiradical potential of these extracts was compared using DPPH and ABTS radical scavenging assays (Table 4). The results obtained using both methods revealed that HG 4:1 extracts from both *Achillea* species possess the most significant antiradical activity (IC_50_ = 0.68% for both species in the DPPH scavenging assay and IC_50_ = 0.30% and 0.38% in the ABTS scavenging assay for *A. millefolium* and *A. biebersteinii* extracts, respectively). In general, the IC_50_ values obtained using the ABTS scavenging method were lower than the IC_50_ values calculated based on the DPPH scavenging analysis. The most significant difference was observed for *A. millefolium* HG 1:1 extract (IC_50_ = 3.58 ± 0.96% in the DPPH scavenging assay and IC_50_ = 0.43 ± 0.14% in the ABTS scavenging assay). The obtained results might be partially explained by the difference in the principles of both assays. Previous studies showed that antiradical activity of hydrophilic compounds is better reflected by the ABTS than the DPPH scavenging assay as the reaction occurs in an aqueous solution [26].

The significant antiradical activity of hydroglycolic *Achillea* extracts results more likely from the content of CQA and DCQA isomers and cynarin as these compounds were previously reported as effective antiradical agents [27,28,29]. The most significant antiradical properties of HG 4:1 extracts might result from the synergistic action of their constituents or antagonistic interactions of the compounds detected in other extracts (eg. HG 1:1). For example, this type of opposite interactions was previously described for 3-CQA (chlorogenic acid), showing synergistic antioxidant activity in the combination with isoquercitrin [30] and antagonistic action in combination with rutin, caffeic acid and rosmarinic acid [31]. Antioxidant properties are very important for cosmetic active ingredients as oxidative stress has been shown to be involved in the premature skin aging, pigmentation disorders and skin carcinogenesis [32,33,34].

The antioxidant potential of *A. millefolium* and *A. biebersteinii* extracts was previously compared by Zengin and co-workers [7]. In this study, water, ethyl acetate and methanolic extracts from flowering plants collected in Turkey were analysed using DPPH and ABTS radical scavenging, reducing power (CUPRAC and FRAP) and metal chelating assays. In the DPPH radical scavenging assay the best results were obtained for methanolic extracts, the highest value being detected for *A. biebersteinii* (126.9 mgTE/g extract). However, in the ABTS radical scavenging assay water extracts were the most effective, with the most significant activity being registered by *A. millefolium* extract (518.1 mgTE/g extract) In the FRAP assay, the highest result was obtained for *A. biebersteinii* methanolic extract (196.12 mg TE/g extract), whereas in the CUPRAC assay, the most potent reducing capacity was shown for *A. millefolium* methanolic extract (255.66 mg TE/g extract) [7]. The protective activity of *A. biebersteinii* extracts from the harmful effect of oxidative stress was also demonstrated in vitro. Pre-treatment of human foreskin fibroblasts (HFF3) with 1 μg/mL methanolic leaf extract was shown to protect the cells from H_2_O_2_ induced toxicity and DNA damage [29].

### 2.3. Tyrosinase Inhibitory Activity

Tyrosinase (EC. 1.14.18.1) as a key enzyme of melanogenesis is a common target of cosmetic active ingredients reducing hyperpigmentation. This cooper-containing enzyme catalyze the rate limiting conversion of L-tyrosine to L-dihydroxyphenylalanine (L-DOPA) (monophenolase activity) and subsequently to dopaquinone (diphenolase activity) [35]. The tyrosinase inhibitory activity of A. millefolium and *A. biebersteinii* hydroglycolyc extracts was compared using both substrates for this enzyme—L-tyrosine and L-DOPA—in order to investigate the influence of the extract compounds on the monophenolase and diphenolase activity of tyrosinase. Obtained results showed that HG 1:1 extracts from *A. millefolium* and *A. biebersteinii* are the most potent tyrosinase inhibitors, decreasing both—the monophenolase and diphenolase activity in all three tested concentrations (5%, 2.5% and 1.25%). Interestingly, HG 4:1 extracts from *A. millefolium* did not influence the tyrosinase activity whereas *A. biebersteinii* extracts inhibited the monophenolase activity by 25–10% in the concentration range from 5–1.25%. Among HG 6:1 extracts only 5% *A. biebersteinii* extract showed signifcant inhibitory activity on the monophenolase activity of tyrosinase (Figure 1).

*A. millefolium* ethyl actetate, methanol and water extracts were shown previously to inhibit monophenolase activity of mushroom tyrosinase with IC_50_ values of 31.57, 23.26 and 15.23 mg KAE/g, respectively. In the same study, the methanol extract from *A. biebersteinii* was found to effectively inhibit monophenolase activity of mushroom tyrosinase, with IC_50_ = 34.24 mg KAE/g. [7]. The extracts described in the mentioned study are however not applicable in cosmetic formulation, except of the water extract, due to the toxicity and irritant potential of the solvents.

Among active compounds identified in the analysed hydroglycolic extract the most probable tyrosinase inhibitors are caffeoylquinic acid (CQA) isomers: 3-CQA, 4-CQA, 5-CQA and a dicaffeoylquinic acid derivative: cynarin (1,3-DCQA). Several CQA isomers were previously described to effectively decrease melanin synthesis in B16F10 murine melanoma cells, inhibit murine tyrosinase in a cell-free assay and downregulate the expression of several genes involved in melanogenesis process, including genes for tyrosinase and tyrosinase-related protein-1 (TRP1) and microphthalmia-associated transcription factor (MITF) [36]. Thus, plant extracts rich in CQA and DCQA isomers may reduce the development of hyperpigmentation disorders by different mechanisms. The highest content of CQA and DCQA isomers was found in *A. biebersteinii* and *A. millefolium* HG 1:1 extracts, showing at the same time the most significant monophenolase and diphenolase inhibitory activity. CQA and DCQA isomers were not detected in *A. millefolium* HG 4:1 and 6:1 extract, showing no tyrosinase inhibitory properties. The correlation between the content of CQA and DCQA isomers and tyrosinase inhibitory activity is also supported by the observation for the HG 4:1 extracts. *A. biebersteinii* HG 4:1 extract, containing significant amounts of 5-CQA (1.435 mg/g), 3-CQA (0.524 mg/g) and 4-CQA (0.209 mg/g) inhibits monophenolase activity of tyrosinase, whereas HG 4:1 extracts from *A. millefolium*, containing mostly quinic acid does not influece tyrosinase activity at tested concentrations.

### 2.4. In Vitro Sun Protection Factor (SPF)

Protection from the harmful effect of the ultraviolet radiation (UVR) is one of most important functions of modern cosmetic products. UVR is responsible for hyperpigmentation disorders and the premature skin aging, known as photoaging [35,37]. It is also considered as the main risk factor for skin cancer [38]. The efficacy of the UV protection of cosmetics is specified by the sun protection factor (SPF), defined as the UV energy required for producing a minimal erythema dose (MED) on protected skin, divided by the UV energy required for producing a MED on unprotected skin. MED is defined as the lowest time interval or dosage of UV irradiation sufficient for producing a minimal, visible erythema on unprotected skin [39]. In addition to several mineral and organic compounds, plant extracts were also described as potent UV filters, effectively increasing the overall SPF of cosmetic products [40]. The SPF values of *A. millefolium* and *A. biebersteinii* HG extracts at 5%, 2.5% and 1.25% were calculated based on the Mansur equation [41]. All analysed extracts showed significant UV-protecting potential, especially at 5% and 2.5% concentrations. The highest SPF values were established for *A. millefolium* HG 4:1 extracts (SPF = 14.04 for 5% and SPF = 7.15 for 2.5%). The lowest SPF was calculated for HG 6:1 extracts with SPF = 9.49 and 8.30 for 5% *A. millefolium* and *A. biebersteinii* extracts, respectively (Table 5). *Achillea* extracts were not previously described in the scientific literature as UV-protecting compounds. However, the essential oil from *A. millefolium* was shown to decrease the activation of MAPK signaling pathway with linalyl acetate as the most active compound [42]. Due to the MAPK activation by UVR, *A. millefolium* might prevent some harmful effects of UVR in the skin.

### 2.5. In Vitro Cytotoxicity

Previously compared biological activities of *A. millefolium* and *A. biebersteinii* HG extracts suggest the potential application of these extracts in the formulation of cosmetics protecting the skin from the undesirable effect of UVR, including oxidative stress and hyperpigmentation disorders. The extracts were then analysed for the potential anti-melanoma activity using human A375 malignant melanoma cells and immortalized keratinocytes HaCaT as a control, noncancerous cell line [43]. As shown in Figure 2 and Figure 3, none of the analysed *A. millefolium* HG extracts was significantly cytotoxic for the melanoma cells within the tested concentration range (0.32–5%). *A. biebersteinii* HG 1:1 and 4:1 extracts decreased the viability of A375 cells at 5%, 2.5% and 1.25% and the observed cytotoxic effect was lower than toward noncancerous cells. It suggests that *A. biebersteinii* HG 1:1 and 4:1 extracts might be potentially useful melanoma-preventing ingredients of topically applied formulations. *A. biebersteinii* HG 6:1 extract at 5% and 2.5% was cytotoxic for A375 melanoma cell and HaCaT keratinocytes at the same level. To our knowledge this is the first report showing potential anti-melanoma activity of *Achillea* extracts.

HaCaT keratinocytes, used as control cells in this study, present several morphological and functional features typical of normal epidermal keratinocytes, making them a good model for the analysis of the irritating potential of novel cosmetic ingredients [44]. Observed cytotoxicity of 5% and 2.5% *A. biebersteinii* HG extracts for human keratinocytes indicate the need for further studies evaluating the safety of their application in cosmetic products. To date, only the data regarding the lack of cytotoxicity of leaf methanol extract from *A. biebersteinii* against human foreskin fibroblasts (HFF3) are available in the scientific literature [14]. Due to the current application of *A. millefolium* extracts in cosmetic formulations the effect of this ingredient on the skin has been studied using various experimental models. Hydro-alcoholic extract from *A. millefolium* was not cytotoxic in vitro for human skin fibroblast cells (HSF-PI-16) and at concentrations <20 mg/mL stimulated their proliferation rate and migratory properties in a scratch wound healing assay [8]. *A. millefolium* extracts at 1% were also not sensitizing in a patch test on participants with atopic dermatitis. No skin irritation was observed also for *A. millefolium* extract in an EpiOcular Human Cell Construct assay (cosmetic product containing 0.00045% of extract) and local lymph node assay (LLNA) using mice (25%, 50% and 100% aqueous extract) [11]. Studies using similar models are necessary in order evaluate the safety of *A. biebersteinii* application as active ingredient of cosmetic formulations.

## 3. Materials and Methods

### 3.1. Chemicals and Reagents

A375 (ATCC CRL-1619) human malignant melanoma cell line was purchased from LGC Standards (Łomianki, Poland). HaCaT immortalized human keratinocytes were purchased from CLS Cell Lines Service GmbH (Eppelheim, Germany). Fetal bovine serum (FBS) was obtained from Pan-Biotech (Aidenbach, Germany). Dulbecco’s modified Eagle’s medium (DMEM)/high glucose, Dulbecco’s phosphate buffered saline (DPBS), mushroom tyrosinase from *Agaricus bisporus*, L-tyrosine, 3,4-dihydroxy-l-phenylalanine (L-DOPA), 2,2-diphenyl-1-picrylhydrazyl (DPPH), 2,2′-azino-bis(3-ethylbenzothiazoline-6-sulfonic acid (ABTS), zinc oxide, L-ascorbic acid, neutral red solution (3.3 g/L) and standards of caffeic acid, chlorogenic acid and quercetin were purchased from Sigma Aldrich (St. Louis, MO, USA). The purity of the reference compounds exceeded 95%. Propylene glycol (>99.8% purity) and K_2_S_2_O_8_ were obtained from Chempur (Piekary Slaskie, Poland). The solvents used for compositional study of the extracts by HPLC-MS, namely water, acetonitrile and formic acid were purchased from Merck (Darmstadt, Germany).

### 3.2. Plant Material and Extraction Procedure

The aerial flowering parts of *Achillea millefolium* and *Achillea biebersteinii* were collected in Kazakhstan, in the Pavlodar region, in the outskirts of Bayanaul village in May 2018. Professor Zuriyadda Sakipova recognized the species, collected the overground parts and dried them in the shade at the temperature not exceeding 30 °C. A voucher specimen of each plant is being kept in the Department of Cosmetology, The University of Information Technology and Management in Rzeszow, Poland with the appropriate identification numbers: KGB2020_3 (*A. millefolium*) and KGB2020_4 (*A. biebersteinii*).

Dried plant material (12 g) was treated with 200 mL water-propylene glycol mixtures in 1:1, 4:1 or 6:1 (*v*/*v*) ratios and subjected to ultrasound-assisted extraction for 30 min using an ultrasonic bath (Sonic-3, Polsonic, Warsaw, Poland). The extraction procedure was repeated three times. The pooled extracts were filtered through the Whatman filter paper, 2 μm pre-filter and a 0.45 μm nylon syringe filter. Hydroglycolic (HG) extracts 1:1, 4:1 and 6:1 were stored at −80 °C until analysis.

### 3.3. The HPLC-ESI-Q-TOF-MS Analysis of the Obtained Hydroglycolic Extracts

The HPLC-ESI-Q-TOF-MS instrumentation was employed for the qualitative and quantitative analyses of both *A. millefolium* and *A. biebersteinii* extracts. Prior to the analysis the solvents were evaporated using Eppendorf Concentrator Plus evaporation system (Thermo Fisher Scientific, Waltham, MA, USA) at 60 °C for 2 days. The following chromatographic and spectrometric conditions were applied to separate, identify and quantify the major metabolites of the samples: the flow rate of 0.2 mL/min, the temperature of 25 °C, the run time of 28 min, and the gradient of acetonitrile with 0.1% formic acid (solvent B) in 0.1% formic acid (A) according to the following scheme 0–1 min—1% B in A, 2 min—8% of B in A, 18 min—60% of B in A, 19–22 min—95% of B in A, 22.10 min—1% of B in A. The mass spectra were recorded in the positive and negative ionization modes within the range of *m*/*z* of 50–1200 u and were handled in the Qualitative Navigator B.08.00 program by Agilent Technologies (Santa Clara, CA, USA). Gas temperature was set at 325 °C, sheath gas at 350 °C, gas flows at 12 L/min, nebulizer at 35 psig, capillary voltage at 3000 V, fragmentor voltage at 100 V, nozzle voltage at 1000 V, and skimmer at 65 V. Each scan provided two MS/MS spectra of the two selected signals with the highest intensity at 10 and 20 V collision energies. Later they were excluded for the following 0.3 min to allow the fragmentation of next signals.

The instrument used in the study, produced by Agilent Technologies, was composed of an HPLC chromatograph 1200 Series (with a binary pump, a degasser, an autosampler, a column thermostat and a PDA detector) and an ESI-Q-TOF-MS mass spectrometer (G6530B). The chromatographic analysis was conducted on a Zorbax Eclipse Plus C-18 column (Agilent Technologies) with dimensions of 150 mm × 2.1. mm and particle size of 3.5 μm.

The quantitative analyses of the selected metabolites were based on the direct comparison of the extracts with the injected reference compounds: caffeic acid, chlorogenic acid and quercetin. For this purpose, the calibration curves of the reference solutions were prepared based on the injections of four different concentrations (0.1 mg/mL, 0.05 mg/mL, 0.025 mg/mL and 0.0125 mg/mL). The following calibration curve equations were obtained: for caffeic acid *y* = 522061991.88*x* + 6759856.23 (R^2^ = 0.9983), for chlorogenic acid *y* = 331271531.54*x* + 10419717.37 (R^2^ = 0.9989) and for quercetin *y* = 618224977.93*x* + 10183935.35 (R^2^ = 0.9857). The quantitative analysis of the selected secondary metabolites present in *Achillea* extracts was prepared based on a direct comparison with the standards which represented a similar class of natural products. The content of cynarin and caffeic acid were calculated from the curve of caffeic acid, the content of kaempferol, jaceidin and axillarin were calculated from the quercetin curve, the content of 3-CQA, 4-CQA, 5-CQA, quinic acid and coumaroylquinic acid isomers were calculated from the chlorogenic acid curve.

### 3.4. Antiradical Activity Assays

#### 3.4.1. DPPH Radical Scavenging Assay

Antioxidant activity of *A. millefolium* and *A. biebersteinii* extracts was assessed using a DPPH radical scavenging assay, according to the procedure described by Matejic et al. [45], with slight modifications. 100 μL of extracts diluted in appropriate solvent (water: propylene glycol 1:1, 4:1 or 6:1) in the concentration range from 100–0.05% was mixed with 100 μL DPPH working solution (25 mM in 99.9% methanol; A_540_ ≈ 1). 100 μL of solvents mixed with 100 μL DPPH were used as a control samples. After 20 min of incubation at room temperature in darkness, the absorbance of the samples was measured at λ = 540 nm using microplate reader (FilterMax F5 Molecular Devices, San Jose, CA, USA). The obtained values of measurements were corrected by the absorbance value of the sample without DPPH. The percentage of DPPH radical scavenging was calculated for each sample based on the following equation:% of DPPH˙ scavenging = [1 − (Abs(S)/Abs(C))] × 100(1)
where: Abs(S)—the corrected absorbance of the extract, Abs(C)—the corrected absorbance of the control sample (DPPH + solvent). The IC_50_ value was defined as the concentration of dried extract in μg/mL that is required to scavenge 50% of DPPH radical activity.

#### 3.4.2. ABTS Radical Scavenging Assay

ABTS radical scavenging assay was preformed according to the protocol described by Re and co-workers [46] with some modifications. 7 mM ABTS solution in 2.45 mM K_2_S_2_O_8_ was prepared and diluted in distilled H_2_O in order to obtain ABTS radical working solution (A_405_ ≈ 1). 15 μL of extracts diluted in appropriate solvent (water: propylene glycol 1:1, 4:1 or 6:1) in the concentration range from 100–0.05% was mixed with 135 μL ABTS working solution. 15 μL of solvents mixed with 135 μL ABTS were used as a control samples. After 15 min of incubation at room temperature in darkness, the absorbance of the samples was measured at λ = 405 nm using microplate reader (FilterMax F5 Molecular Devices). The obtained values of measurements were corrected by the absorbance value of the sample without ABTS. The percentage of ABTS radical neutralization was calculated based on the following equation:% of ABTS scavenging = [1 − (Abs(S)/Abs(C))] × 100(2)
where: Abs(S)—the corrected absorbance of the extract, Abs(C)—the corrected absorbance of the control sample (ABTS + solvent). The IC_50_ value was defined as the concentration of dried extract in μg/mL that is required to scavenge 50% of ABTS radical activity.

### 3.5. Mushroom Tyrosinase Inhibitory Assay

The inhibition of the monophenolase activity of mushroom tyrosinase by *Achillea* HG extracts was performed based on the protocol described by Wang and co-workers [47] with some modifications. 80 μL of phosphate buffer (100 mM, pH = 6.8) was mixed with 20 μL of diluted sample and 20 μL mushroom tyrosinase (500 U) and pre-incubated for 10 min at room temperature. Next, 80 μL 2 mM L-tyrosine was added and the samples were incubated for further 20 min at room temperature, in darkness. The inhibition of the diphenolase activity of tyrosinase was determined based on the method described by Uchida et al. [48]. Phosphate buffer (120 μL, 100 mM, pH = 6.8) was mixed with 20 μL of diluted sample and 20 μL mushroom tyrosinase (500 U) and pre-incubated for 10 min at room temperature. Next, 40 μL 4 mM L-DOPA was added and the samples were incubated for further 20 min at room temperature, in darkness. In both experiment procedures the formation of dopachrome was measured spectrophotometrically at λ = 450 nm using FilterMax F5 microplate reader (Molecular Devices). The obtained values were corrected by the absorbance of the extracts without tyrosinase and the substrate (L-tyrosine or L-DOPA). Control sample (100% tyrosinase activity) contained phosphate buffer, tyrosinase, equal volume of the solvent and the appropriate dose of the substrate. 100 μg/mL kojic acid was used as a reference tyrosinase inhibitor. The activity of tyrosinase was calculated based on the equation:% of tyrosinase activity = [Abs(S)/Abs(C)] × 100%(3)
where: Abs(S)—the absorbance of the sample (extract + tyrosinase + substrate), Abs(C)—the absorbance of the control sample (solvent + tyrosinase + substrate).

### 3.6. In Vitro Cytotoxicity Assay

The cytotoxic effect of HG *Achillea* extracts was evaluated using Neutral Red Uptake Test [49]. A375 human melanoma and HaCaT human keratinocyte cell lines were maintained in DMEM supplemented with 10% FBS at 37 °C in a humidified atmosphere with 5% CO_2_. For the experimental purpose the cells were plated onto a 96-well plate (3000 per well) and grown overnight. The cells were then treated with various concentrations of HG *Achillea* extracts (0–5%) or equal volumes of appropriate solvents as controls. Following 48 h of culture the cells were incubated for 3 h in culture medium containing 33 μg/mL neutral red and rinsed with DPBS. The morphology of the cells was examined using Nikon Eclipse inverted microscope (Nikon, Tokyo, Japan) equipped with Invenio II 5S camera (DeltaPix, Smorum, Denmark). The cells were lysed using acidified ethanol solution (50% *v*/*v* ethanol, 1% *v*/*v* acetic acid). The absorbance of the released neutral red was measure using FilterMax F5 microplate reader (Molecular Devices) at λ = 540 nm and corrected by the absorbance at λ = 620 nm. Mean absorbance value of the lysate from cells cultures in the presence of the solvent was set as 100% cellular viability and used to calculate the percentage of viable cells following extracts treatment.

### 3.7. Determination of the In Vitro Sun Protection Factor

*In vitro* sun protection factor (SPF) was calculated based on the measurements of the absorbance of 5%, 2.5% and 1.25% *Achillea* HG extracts within the wavelength range from 290–320 nm obtained using DR600 UV-Vis spectrophotometer (Hach Lange, Wrocław, Poland). The solvents (mixtures of distilled water and propylene glycol in 1:1, 4:1 or 6:1 radios) were used as blank samples. For SPF determination the Mansur Equation (4) [41] was applied and EE × I values determined by Sayre [50] (Table 6) were used:(4)$$SPF=CF\times \overset{320}{\underset{290}{\sum }}EE\;\left(\lambda \right)\times I\;\left(\lambda \right)\times Abs\;\left(\lambda \right)$$
where: EE (λ)—erythemal effect spectrum; I (λ)—solar intensity spectrum; Abs (λ)—absorbance of the sample; CF—correction factor (=10).

### 3.8. Statistical Analysis

All experiments were conducted in at least three replicates. Obtained data were analyzed using GraphPad Prism 7.0 Software (GraphPad Software, San Diego, CA, USA) and Statistica 13.0 Software (StatSoft, Kraków, Poland). The statistical significance between results obtained for different extracts were analyzed using one-way, two-way or three-way ANOVA followed by Tukey’s test. All data are showed as mean ±SD.

## 4. Conclusions

This study presents an evaluation of potential cosmetic application of hydroglycolic extracts from the flowering aerial parts of *A. biebersteinii* in comparison with hydroglycolic extracts of *A. millefolium,* a species currently used in cosmetic formulations. Application of various H_2_O:polyethylene glycol ratios (1:1, 4:1 and 6:1, *v*/*v*) during the extraction process allowed us to optimize the extraction procedure. Qualitative and quantitative analysis showed that the extracts obtained from *A. biebersteinii* are characterized by more diverse phytochemical composition than the extracts obtained from *A. millefolium*. The most abundant bioactive compounds found in *A. biebersteinii* extracts were CQA and DCQA isomers (cynarin, 2-CQA, 4-CQA and 5-CQA). The results of the conducted biological activity studies strongly support the application of *A. bieberstinii* hydroglycolic extracts in cosmetic products as they display significant antiradical, tyrosinase inhibitory and sun protective properties. In contrast with *A. millefolium* extracts, *A. biebersteinii* HG 1:1 and 4:1 extracts at 2.5% were also cytotoxic against the human malignant melanoma cell line A375, showing lower cytotoxicity against noncancerous HaCaT human keratinocytes. Conducted research showed that *A. biebersteinii* HG 1:1 and 4:1 extracts possess several activities important for active ingredients of cosmetics, especially formulations protecting the skin from the harmful effects of UVR. The extracts might also be valuable multifunctional ingredients for anti-pollution cosmetics, a newly developed type of cosmetic products designed to protect the skin from the adverse impact of air pollutants and climatic factors [51]. However, the safety and irritating potential of *A. biebersteinii* hydro-glycolic extracts requires further experimental verification, for example by using patch test on human volunteers or 3D epidermal skin models.

## Figures and Tables

**Figure 1 molecules-25-03368-f001:**
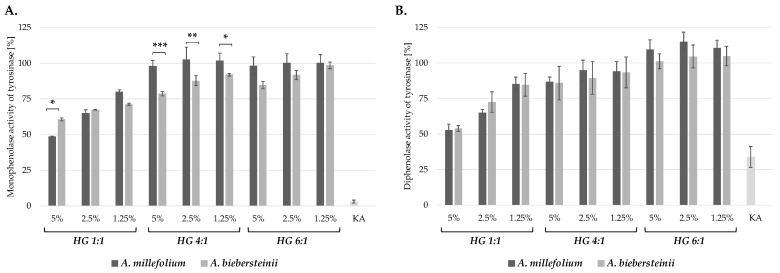
Inhibitory effect of *A. millefolium* and *A. biebersteinii* extracts on the monophenolase (**A**) and diphenolase (**B**) activity of mushroom tyrosinase in comparison with 50 μg/mL kojic acid (KA); values on graphs represent mean ± SD (*n = 3*); * *p* < 0.05, ** *p* < 0.001, *** *p* < 0.0001.

**Figure 2 molecules-25-03368-f002:**
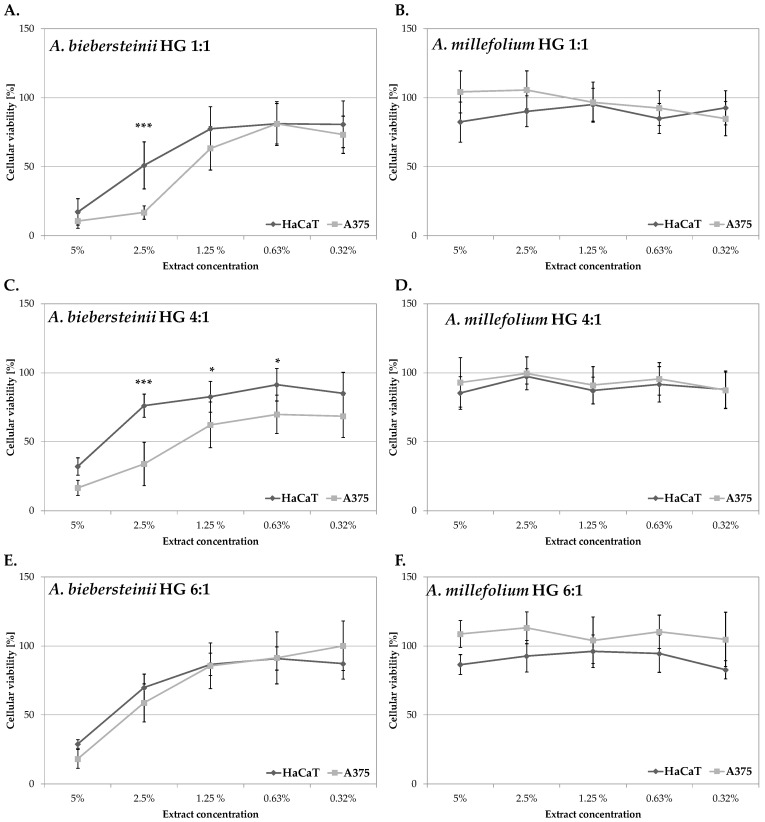
In vitro cytotoxicity of *A. biebersteinii* (**A**,**C**,**E**) and *A. millefolium* (**B**,**D**,**F**) extracts against human keratinocytes HaCaT and A375 human malignant melanoma cell lines; values on graphs represent mean ± SD (*n = 3*); * *p* < 0.05, *** *p* < 0.0001.

**Figure 3 molecules-25-03368-f003:**
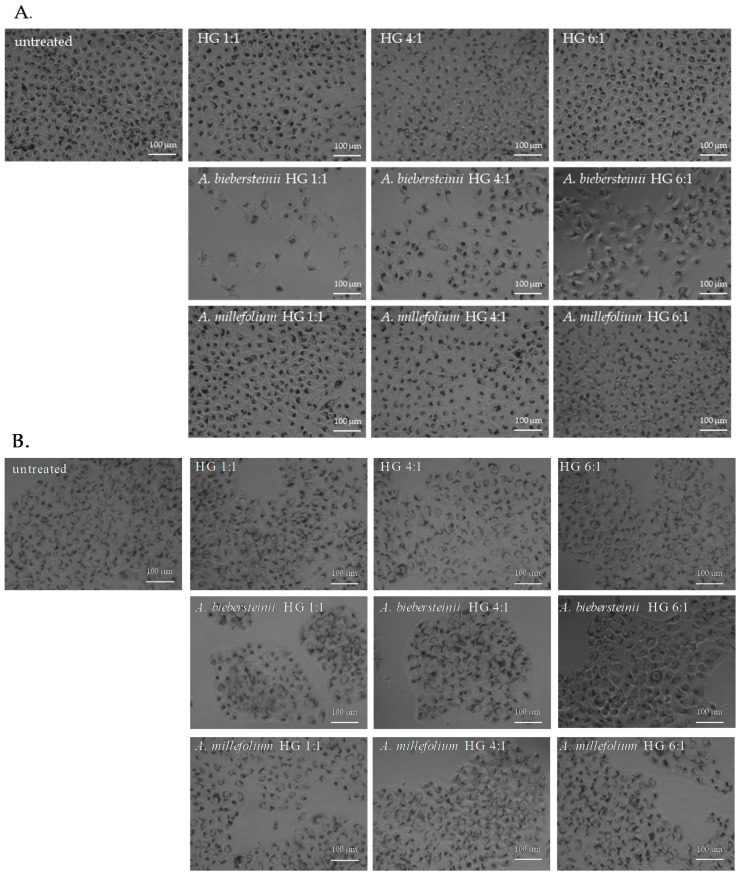
Morphology of A375 (**A**) and HaCaT (**B**) cells treated for 48 h with tested extracts or appropriate solvents at 2.5% concentration; the cells were stained with neutral red; pictures are representative for three experiments, magnification 4×, scale bar = 100 μm.

**Table 1 molecules-25-03368-t001:** The tentatively identified compounds in *Achillea biebersteinii* extracts.

Ionization Mode	Rt [min]	Molecular Formula	*m*/*z* Experimental	*m*/*z* Calculated	Delta [ppm]	DBE	Tentative Compound	MS/MS Fragments	Ref.	AB HG 1:1	AB HG 4:1	AB HG 6:1
[M−H]^−^	4.2	C_7_H_12_O_6_	191.0573	191.0561	−6.19	2	Quinic acid	129, 101	[21]	+	+	-
[M−H]^−^	8.5	C_16_H_18_O_9_	353.0917	353.0878	−11	8	3-caffeoylquinic acid	191, 179	[7,19]	+	+	-
[M−H]^−^	10.6	C_16_H_18_O_9_	353.0914	353.0878	−10.15	8	5-caffeoylquinic acid	191, 179	[22]	+	+	-
[M−H]^−^	11.4	C_16_H_18_O_9_	353.0916	353.0878	−10.72	8	4-caffeoylquinic acid	191	[7,19]	+	-	-
[M−H]^−^	11.5	C_9_H_8_O_4_	179.035	179.035	−0.1	6	Caffeic acid	135	[19]	+	-	+
[M−H]^−^	11.7	C_16_H_18_O_8_	337.0918	337.0929	3.23	8	Coumaroyl-quinic acid isomers	191,173	[18,19]	+	-	-
[M−H]^−^	12.6	C_16_H_18_O_8_	337.0964	337.0929	−10.67	8	Coumaroyl-quinic acid isomers	191	[18,19]	+	+	-
[M−H]^−^	14.8	C_25_H_24_O_12_	515.1195	515.1195	0	14	Cynarin	353, 191, 179	[7,19]	+	+	+
[M−H]^−^	17.5	C_15_H_10_O_6_	285.0407	285.0405	−0.83	11	Kaempferol	133	[23]	+	+	+
[M−H]^−^	18.2	C_17_H_14_O_8_	345.0604	345.0616	3.44	11	Axillarin	330, 315	[18]	+	+	+
[M−H]^−^	20.4	C_17_H_14_0_7_	329.065	329.0667	5.08	11	3,8-Dimethylherbacetin	314, 299	[18]	+	+	-
[M−H]^−^	20.6	C_18_H_16_O_8_	359.076	359.0772	3.45	11	Jaceidin	329, 344	[19]	+	+	+

Rt—retention time, Delta—difference between experimental and calculated mass (mmu), DBE-double bond equivalent, Ref—references, + detected, - not detected, AM—*Achillea millefolium*, AB—*Achillea biebersteinii*, HG—hydroglycolic extract.

**Table 2 molecules-25-03368-t002:** Tentatively identified compounds in *Achillea millefolium* extracts.

Ionization Mode	Rt [min]	Molecular Formula	*m*/*z* Experimental	*m*/*z* Calculated	Delta [ppm]	DBE	Tentative Compound	MS/MS Fragments	Ref.	AM HG 1:1	AM HG 4:1	AM HG 6:1
[M−H]^−^	3.9	C_7_H_12_O_6_	191.0568	191.0561	−3.58	2	Quinic acid	129, 101	[21]	+	+	+
[M−H]^−^	8.4	C_16_H_18_O_9_	353.091	353.0878	−9.02	8	3-caffeoylquinic acid	191, 179	[7,19]	+	-	-
[M−H]^−^	10.3	C_16_H_18_O_9_	353.0905	353.0878	−7.61	8	5-caffeoylquinic acid	191, 179	[22]	+	+	-
[M−H]^−^	11.3	C_16_H_18_O_9_	353.0904	353.0878	−7.33	8	4-caffeoylquinic acid	191, 179	[7,19]	+	-	-
[M−H]^−^	11.7	C_16_H_18_O_8_	337.0958	337.0929	−8.6	8	Coumaroyl-quinic acid isomers	191	[18,19]	+	-	-
[M−H]^−^	12.56	C_16_H_18_O_8_	337.0958	337.0929	−8.6	8	Coumaroyl-quinic acid isomers	191	[9,19]	+	-	-
[M−H]^−^	14.8	C_25_H_24_O_12_	515.1243	515.1195	−9.3	14	Cynarin	353, 179	[7,19]	+	-	-
[M−H]^−^	17.5	C_15_H_10_O_6_	285.0386	285.0405	6.51	11	Kaempferol	193, 127	[23]	+	+	+
[M−H]^−^	20.6	C_18_H_16_O_8_	359.0769	359.0772	0.95	11	Jaceidin	329, 344	[18]	+	-	-

Rt—retention time, Delta—difference between experimental and calculated mass (mmu), DBE-double bond equivalent, Ref—references, + detected, - not detected, AM—*Achillea millefolium*, AB—*Achillea biebersteinii*, HG—hydroglycolic extract.

**Table 3 molecules-25-03368-t003:** Quantitative analysis of the selected identified compounds in *A. millefolium* and *A. biebersteinii* hydroglycolic extracts *; each value represents mean content of the compound in mg per 100 g of dried powder ± SD (*n = 3*); ND—not detected**.**

Compound/Extract	*Achillea biebersteinii*			*Achillea millefolium*		
	HG 1:1	HG 4:1	HG 6:1	HG 1:1	HG 4:1	HG 6:1
Cynarin	1.215 ± 0.008	0.076 ± 0.004 ^a^	0.078 ± 0.000 ^a^	0.119 ± 0.002	ND	ND
3-caffeoylquinic acid	1.922 ± 0.089	0.524 ± 0.011 ^b^	0.451 ± 0.039 ^b^	0.595 ± 0.068 ^b^	ND	ND
5-caffeoylquinic acid	3.181 ± 0.139	1.435 ± 0.033	1.102 ± 0.093 ^c^	0.993 ± 0.092 ^c^	0.036 ± 0.004	ND
4-caffeoylquinic acid	1.207 ± 0.021	0.209 ± 0.024 ^d^	0.193 ± 0.005 ^d^	0.466 ± 0.013	ND	ND
Caffeic acid	0.036 ± 0.003^e^	0.035 ± 0.009 ^e^	0.037 ± 0.005 ^e^	ND	ND	ND
Kaempferol	0.784 ± 0.001	0.141 ± 0.003	0.113 ± 0.012	0.272 ± 0.021	0.065 ± 0.001	0.022 ± 0.001
Jaceidin	0.815 ± 0.048	ND	0.005 ± 0.001 ^f^	0.043 ± 0.008 ^f^	ND	ND
Axillarin	2.539 ± 0.265	0.055 ± 0.007	ND	ND	ND	ND
Coumaroyl-quinic acid isomers	0.203 ± 0.016 ^g^	0.129 ± 0.010 ^g^	0.131 ± 0.000 ^g^	0.734 ± 0.069	ND	ND
Coumaroyl-quinic acid isomers	0.121 ± 0.007 ^h^	0.106 ± 0.002 ^h^	0.117 ± 0.012 ^h^	1.733 ± 0.105	ND	ND
Quinic acid	0.684 ± 0.470 ^i^	0.868 ± 0.008 ^i,j^	0.071 ± 0.000	1.172 ± 0.142 ^i,k^	1.201 ± 0.050 ^i,k^	1.294 ± 0.126 ^j,k^
3,8-Dimethylherbacetin	0.660 ± 0.025	0.023 ± 0.002	ND	ND	ND	ND

* The contents of cynarin and caffeic acid were calculated from the curve of caffeic acid, the content of kaempferol, jaceidin and axillarin was obtained from the quercetin curve, the concentration of 3-CQA, 4-CQA, 5-CQA, quinic acid and coumaroylquinic acid isomers were calculated from the chlorogenic acid curve. ^a–k^ Means in each row not sharing the same letter are significantly different at *p* < 0.05.

**Table 4 molecules-25-03368-t004:** DPPH and ABTS radical scavenging activity of hydroglycolic (HG) extracts from *A. millefolium* and *A. biebersteinii;* each value represents mean ± SD (*n = 3*). Means in each column not sharing the same letter are significantly different at *p* < 0.05.

	IC_50_ (%) ± SD		
		DPPH Scavenging	ABTS Scavenging
*A. millefolium*	HG 1:1	3.58 ± 0.96	0.43 ± 0.14 ^b,c^
	HG 4:1	0.68 ± 0.02 ^a^	0.30 ± 0.03 ^b^
	HG 6:1	1.68 ± 0.38 ^a^	0.49 ± 0.14 ^b,c^
*A. biebersteinii*	HG 1:1	0.91 ± 0.05 ^a^	0.72 ± 0.17 ^c^
	HG 4:1	0.68 ± 0.05 ^a^	0.38 ± 0.04 ^b^
	HG 6:1	1.01 ± 0.08 ^a^	0.56 ± 0.05 ^b,c^
*Vitamin C*		0.78 ± 0.05 μg/mL	0.46 ± 0.02 μg/mL

^a–c^ Means in each column not sharing the same letter are significantly different at *p* < 0.05.

**Table 5 molecules-25-03368-t005:** In vitro sun protection factor (SPF) of hydroglycolic (HG) extracts from *A. millefolium* and *A. biebersteinii*, each value represents mean ± SD (*n = 3*).

		5%	2.5%	1.25%
*A. millefolium*	HG 1:1	12.24 ± 0.20	5.65 ± 0.15 ^b^	2.59 ± 0.21 ^d,f^
	HG 4:1	14.04 ± 0.17	7.15 ± 0.14	2.98 ± 0.18 ^f^
	HG 6:1	9.49 ± 0.24	4.78 ± 0.08	1.90 ± 0.11 ^c,e^
*A. biebersteinii*	HG 1:1	11.67 ± 0.30 ^a^	5.74 ± 0.13 ^b^	2.24 ± 0.06 ^c,d,e,f^
	HG 4:1	11.64 ± 0.09 ^a^	5.90 ± 0.14 ^b^	2.37 ± 0.13 ^c,d^
	HG 6:1	8.30 ± 0.04	3.86 ± 0.16	1.85 ± 0.21 ^e^
*Zinc oxide*	1 mg/mL	16.79 ± 0.49		

^a–f^ Means in table not sharing the same letter are significantly different at *p* < 0.05.

**Table 6 molecules-25-03368-t006:** Normalized product function used in the calculation of SPF. EE—erythremal effect spectrum, I—solar intensity spectrum; values adapted from the work of Sayre and co-workers [50].

Wavelength (λ, nm)	EE × I (Normalized)
290	0.0150
295	0.0817
300	0.2874
305	0.3278
310	0.1864
315	0.0839
320	0.0180
Total	1

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
