# Peer review of "Achillea millefolium L. and Achillea biebersteinii Afan. Hydroglycolic Extracts–Bioactive Ingredients for Cosmetic Use"

_molecules, 2020, doi:10.3390/molecules25153368_

Round 1

Reviewer 1 Report

The contribution molecules-868107 presents a study on Achillea millefolium L. and Achillea biebersteinii Afan hydroglycolic extracts bioactive ingredients for cosmetic use. Overall, the manuscript discusses Achillea millefolium L. and Achillea biebersteinii Afan hydroglycolic extracts characteristics and properties. However, the authors do not include enough discussion about the structure-function relationship between the extracts compounds and bioactive properties.

The following suggestions may improve the manuscript:

The introduction can be enhanced by adding some references concerning examples of bioactive ingredients for cosmetics.

Section 2.2 must be improved. Why only DPPH test was used? Why was 4:1 extract more effective in DPPH radical scavenging than HG 1:1 and 6:1 extracts?

In section 2.7 some cell images would be helpful to show the potential anti-melanoma activity of Achillea extracts.

The conclusion section must be improved. A conclusion is not an abstract of the manuscript; it is a synthesis of key points. It should answer the question posed by the authors.

Reviewer 2 Report

This manuscript describes phytochemical composition and quantitation for hydroglycolic extracts from two Achillea species, A. millefolium and A. biebersteinii, using HPLC-ESI-Q-TOF-MS, and in vitro biological activities such as DPPH radical scavenging, tyrosinase inhibition, cytotoxicity and sun protection factor relating to their cosmetic use. A few scientific investigations of the plant Achillea biebersteinii were reported, and comparative analysis of hydroglycolic extracts from two plants A. millefolium and A. biebersteinii seem to be interesting topic. However, the manuscript has important flaws in the method design and data interpretation which are not acceptable.

1. For the quantitative analysis for identified compounds in plant extracts using HPLC-ESI-QTOF-MS, the authors got calibration curves using references having similar structures or scaffolds, not exact reference compounds. Basically, it is not valid.

2. Some compound names or abbreviations are not reasonable: what are 5-DCQA, 3-DCQA, 4-DCQA?? Dicaffeoylquinic acid? They are 5-caffeoylquinic acid, 3-caffeoylquinic acid, 4-caffeoylquinic acid as shown in Tables 2 and 3? Also, the compound name 5,7,4-trihydroxy-3,6-dimethoxy flavonone is invalid. “Flavonone” maybe misspelling of flavanone or flavone. In addition, the reported compounds (isoflavones) in a reference [27] are not the identical with this compound (page 7).

3. The hydroglycolic solvents (water-propylene glycol mixtures) were used for extraction, and it could not be removed. Thus, all the extracts in this paper contain water-propylene glycol mixture. However, the authors did not show effects of the corresponding solvents in the assay systems (e.g. controls with/without hydroglycolic solvents).

4. In the Figure S3, the content of identified compounds has been described as mg/100g dried extract (Y axis). So, the hydroglycolic solvents in the samples used were completely removed?

Reviewer 3 Report

The authors report on the phytochemical characterization (by LC-ESI-Q-TOF-MS) as well as screening for in vitro biological activities (DPPH radical scavenging activities, tyrosinase, sun protection factor) and cytotoxicity (on 2 human cell lines) of extracts from 2 Achillea species (A. millefolium and A. biebersteinii). Extracts were prepared in agreement with green chemistry and cosmetic requirements.

The paper is well written, concise and the results well discussed. New results about the phytochemistry and biological activities of the extracts are presented. This paper deserve publication in Molecules following few minor corrections:

1- Abstract:  avoid the use of abbreviations (at least without defining them before). E.g.: 3-DCQA, 5-DCQA, SPF

2- Keywords: provide complete meaning of: SPF, HPLC-Q-TOF-MS

3- Both HPLC-ESI-Q-TOF-MS and HPLC-Q-TOF-MS are used as abbreviations. Please choose one.

4- As for tyrosinase assay, please provide positive control value for DPPH radical scavenging activity and SPF (from your own measurements (if possible, it would be better) or at least from the literature) for comparison.

5- Plant species have to be in italics (e.g. A. millefolium lines 207, 2014, 228, 230, 234…; A. bierbersteinii lines 228, 236…).

6- L-DOPA, L-tyrosine: “L” in small capital letter.

7- Please provide statistical analyses for Tables 3, 4 and 5.

Round 2

Reviewer 1 Report

The authors have addressed all my suggestions.

I found their responses satisfactory and the revised version has been much improved.

My last comment is that Figure 3 is not cited within the main text.

Reviewer 2 Report

The manuscript shows phytochemical composition and in vitro biological activities of hydroglycolic extracts derived from Achillea millefolium and A. biebersteinii for cosmetic application. The revised version has been improved considerably enough to be published, however, there are still some typographical errors found in the main text and supplementary data. Thus, the reviewer recommends to check English language and compound names etc.